# mTOR: A Potential New Target in Nonalcoholic Fatty Liver Disease

**DOI:** 10.3390/ijms23169196

**Published:** 2022-08-16

**Authors:** Jiayao Feng, Shuting Qiu, Shipeng Zhou, Yue Tan, Yan Bai, Hua Cao, Jiao Guo, Zhengquan Su

**Affiliations:** 1Guangdong Engineering Research Center of Natural Products and New Drugs, Guangdong Provincial University Engineering Technology Research Center of Natural Products and Drugs, Guangdong Pharmaceutical University, Guangzhou 510006, China; 2Guangdong Metabolic Disease Research Center of Integrated Chinese and Western Medicine, Key Laboratory of Glucolipid Metabolic Disorder, Ministry of Education of China, Guangdong TCM Key Laboratory for Metabolic Diseases, Guangdong Pharmaceutical University, Guangzhou 510006, China; 3School of Public Health, Guangdong Pharmaceutical University, Guangzhou 510310, China; 4School of Chemistry and Chemical Engineering, Guangdong Pharmaceutical University, Zhongshan 528458, China

**Keywords:** mTOR, nonalcoholic fatty liver disease, targeted therapy, notch, hedgehog, hippo

## Abstract

The global prevalence of nonalcoholic fatty liver disease (NAFLD) continues to rise, yet effective treatments are lacking due to the complex pathogenesis of this disease. Although recent research has provided evidence for the “multiple strikes” theory, the classic “two strikes” theory has not been overturned. Therefore, there is a crucial need to identify multiple targets in NAFLD pathogenesis for the development of diagnostic markers and targeted therapeutics. Since its discovery, the mechanistic target of rapamycin (mTOR) has been recognized as the central node of a network that regulates cell growth and development and is closely related to liver lipid metabolism and other processes. This paper will explore the mechanisms by which mTOR regulates lipid metabolism (SREBPs), insulin resistance (Foxo1, Lipin1), oxidative stress (PIG3, p53, JNK), intestinal microbiota (TLRs), autophagy, inflammation, genetic polymorphisms, and epigenetics in NAFLD. The specific influence of mTOR on NAFLD was hypothesized to be divided into micro regulation (the mechanism of mTOR’s influence on NAFLD factors) and macro mediation (the relationship between various influencing factors) to summarize the influence of mTOR on the developmental process of NAFLD, and prove the importance of mTOR as an influencing factor of NAFLD regarding multiple aspects. The effects of crosstalk between mTOR and its upstream regulators, Notch, Hedgehog, and Hippo, on the occurrence and development of NAFLD-associated hepatocellular carcinoma are also summarized. This analysis will hopefully support the development of diagnostic markers and new therapeutic targets in NAFLD.

## 1. Introduction

Nonalcoholic fatty liver disease (NAFLD) affects 24% of the global population [1]. NAFLD ranges from nonalcoholic fatty liver (NAFL) to nonalcoholic steatohepatitis (NASH), which can progress to cirrhosis and hepatocellular carcinoma (HCC) [2,3]. Since the discovery of NAFLD, its pathogenesis has remained unclear. At present, the classical “second strike” hypothesis advocates that lipid aggregation in the cytoplasm of liver cells (the first strike) triggers a series of cytotoxic events (the second strike), leading to inflammatory responses in the liver. The main influencing factors of NAFLD include insulin resistance (IR), oxidative stress (OS), which is caused by the considerable production of free radicals [4], and inflammatory responses induced by reactive oxygen species (ROS) [5] in liver parenchymal cells. Additional research revealed that NAFLD is a liver disease related to genetics, the environment, metabolism, stress, and inflammation, thus producing the “multiple strikes” theory. This theory focuses on the role of the gut (intestinal microbiome) in liver axis, autophagy, inflammation, inherited genetic polymorphisms, and genetic adaptations [6,7].

Therefore, data have shown that NAFLD is driven by metabolic syndrome and is closely related to lipid metabolism [via sterol regulatory element-binding proteins (SREBPs) [8], IR {via forkhead box O1 (Foxo1) and phosphatidate phosphatase 1 (Lipin-1)}, {OS via p53-induced gene 3 (PIG3), p53, c-Jun N-terminal kinase (JNK)}, the intestinal flora {via toll-like receptor (TLRs)}, autophagy, inflammation, genetic polymorphisms, and epigenetics [9,10]. To date, requirements from some phase 2 and phase 3 clinical trials for NAFLD have not been met and have barely met the currently required histological criteria [11]. These apparent failures partly reflect the current lack of precise disease and target definitions, as well as the incomplete understanding of the disease pathogenesis; therefore, there is an urgent need to identify key targets or pathways that affect the development and progression of NAFLD.

Studies have found that the mechanistic target of rapamycin (mTOR) inhibitors can block the mTOR signal transduction pathway, produce anti-inflammatory and anti-proliferative effects in cancer, and induce autophagy and apoptosis; thus, these agents are being used in the treatment of NAFLD-associated HCC and other cancers [12]. The target of rapamycin (TOR) kinase is commonly known as mTOR [13]. Rapamycin, which was first identified and named by Sehgal and colleagues, is an antifungal antibiotic produced by Streptomyces bibulus NRRL 5491, which was subsequently discovered in mammalian cells. mTOR is a major regulator of cell growth and development that senses and integrates various nutritional and environmental cues [14]. Since its discovery, mTOR has been recognized as the central node of a network that controls cell growth and development and regulates liver lipid metabolism through various mechanisms [15]. However, the specific impact of mTOR on NAFLD is not clear, so this paper will start from mTOR. In this paper, the specific mechanism of mTOR on NAFLD and the relationship between each of the influencing factors are reviewed regarding lipid metabolism (SREBPs), insulin resistance (Foxo1, Lipin1), oxidative stress (PIG3/P53/JNK), intestinal microbiota (TLRs), autophagy, inflammation, genetic polymorphism, and crown genetics. In this paper, the specific influence of mTOR on NAFLD was hypothesized to be divided into micro regulation (the mechanism of influence of mTOR on NAFLD factors) and macro mediation (the relationship between various influencing factors) to summarize the influence of mTOR on the development process of NAFLD and prove the importance of mTOR in the influencing factors of NAFLD regarding multiple aspects. The effects of crosstalk between mTOR and its upstream regulators, Notch, Hedgehog, and Hippo, on the occurrence and development of NAFLD-associated HCC are also summarized. This analysis will hopefully support the identification and development of diagnostic markers and new therapeutic targets in NAFLD.

## 2. Architecture of mTORC1 and mTORC2

mTOR is a 289 kDa serine/threonine protein kinase in the phosphatidylinositol 3-kinase-related kinase (PIKK) family. mTOR plays a central role in the signaling pathways involved in the control of cell growth and proliferation and exists in two complexes: mTORC1 and mTORC2. It is currently believed that mTORC1 plays a more important role than mTORC2 and that the mTORC2 signaling pathway is relatively simple. However, in general, mTOR signaling can promote substance metabolism, participate in apoptosis and autophagy, and play important roles in a variety of diseases [16]. mTORC1 is composed primarily of mTOR, a regulatory-associated protein of mTOR (Raptor), mammalian lethal with SEC13 protein 8 (mLST8), proline-rich Akt substrate protein of 40 kD (PRAS40), and DEP domain-containing mTOR-interacting protein (Deptor) [17] (Figure 1). The mTOR core is a complex protein comprising HEAT repeats [18], a FAT domain, an FRB domain, a kinase domain, and a FATC domain, as determined by cryogenic electron microscopy and crystallography analysis. PRAS40 [19] and Raptor [20] can act together as endogenous inhibitors of mTORC1. PRAS40 is a target for insulin regulation by mTORC1 activity inhibitors [21]. The FRB domain is the main target of 13 kDa FK506- and rapamycin-binding protein (FKBP2) and is inhibited by this protein, which explains why mTORC1 is sensitive to rapamycin [22]. mLST8 can directly act on the kinase domain to stabilize mTORC1 [23].

mTORC2 [24] has the same mTOR, mLST8, and Deptor [25] structures as mTORC1. However, it differs due to the presence of a rapamycin-insensitive chaperone of mTOR (Rictor), the stress-activated map kinase-interacting protein (mSIN1), and a protein observed with Rictor 1/2 (Protor1/2) [26]. As described above, mTORC2 and mTORC1 harbor the same crystallographic mTOR structure [25], and mLST8 [23] is crucial for mTORC2 structural stability. The main difference between mTORC2 and mTORC1 lies in the Rictor scaffold protein, which blocks the effect of mSIN1 on the main mTOR structure and the FKBP12 binding site on mTOR; thus, mTORC2 [27] is rendered insensitive to acute inhibition by rapamycin. These findings have been verified by Li Ting et al. [28], who found that the rapamycin-sensitive mTORC1 can regulate cell proliferation, apoptosis, and autophagy and affect lipid metabolism. mTORC2 is mainly involved in cytoskeletal construction and cell motility and has some degree of resistance to rapamycin (Figure 1).

## 3. mTOR Regulates Liver Lipid Metabolism through SREBPs

It is well known that the liver is the central organ for lipid oxidation, fat synthesis, and lipid metabolism. The main cause of NAFLD is an imbalance in the supply, synthesis, consumption, and clearance of triglycerides in the liver, which results in liver fat deposition [29]. Studies have shown that in addition to reducing the expression of SREBP target genes related to liver lipid metabolism, rapamycin can also reduce the levels of acetyl-coenzyme A carboxylase (ACC) [30], fatty acid synthase (FASN) [31], and stearoyl-CoA desaturase 1 (SCD-1) [32] and is an inactive precursor in the endoplasmic reticulum that requires proteolysis to translocate from the endoplasmic reticulum to the nucleus; previous studies confirmed that this process is regulated by mTORC1. Moreover, the most recent study on this topic found that the treatment of cells with palmitic acid to simulate intracellular lipid toxicity led to the activation of an important stimulator of the interferon genes (STING) pathway, which can regulate the expression of SREBPs and further activate the TBK1—p62 axis to increase mTORC1 enzyme activity, thereby significantly inhibiting the degradation of intracellular lipid droplets. Meanwhile, analyses of human fatty liver tissue specimens confirmed that the degree of STING/mTORC1 activation was highly correlated with the level of liver inflammation and positively correlated with the number of lipid droplets in liver cells [33]. It has been hypothesized that targeting STING/mTORC1 may be a new approach to inhibit chronic inflammation in fatty livers and reduce fatty acid accumulation in liver cells. In addition, recent research has confirmed that the GTPase-activating protein, folliculin (FLCN), can selectively inhibit mTORC1 function, resulting in the altered intracellular localization of transcription factor binding to IGHM enhancer 3 (TFE3). After TFE3 enters the nucleus, INSIG2 expression increases, consequently inhibiting SREBP-1c activity and decreasing the expression of genes related to lipid synthesis. In addition, FLCN knockdown can increase the expression of lipid oxidation-related genes in a TFE3-dependent manner, which reduces lipid accumulation in the liver in two ways and alleviates the progression of NAFLD [34]. These data provide insight into the mechanism by which mTORC1 regulates liver lipid homeostasis. The above approaches are discussed with a focus on the mTOR-mediated control of SREBP expression to regulate lipid accumulation and inflammation in NAFLD.

In the nucleus, mTORC1 indirectly regulates the transcription of SREBP-1c and de novo lipogenesis (DNL) genes, thereby affecting liver lipid metabolism and fat deposition. Portsmann and colleagues were the first to discover, in 2008, that rapamycin negatively impacts the nuclear accumulation of SREBPs and downregulates the expression of DNL genes [35]. Subsequently, in 2010 [36] and 2011 [37], Manning’s laboratory found that mTORC1 could somewhat regulate SREBP-1c activity and that increased DNL gene expression could serve as a marker of HCC cell proliferation [38]. However, SREBPs and DNL-related genes are related to lipid metabolism, and recent studies on genes related to lipid metabolism and mTOR were not comprehensive, perhaps because these researchers limited their analyses to extensively studied genes, such as SREBPs and DNL-related genes. Future studies on lesser understood genes involved in lipid metabolism will hopefully confirm the relationship among lipid metabolism-related genes, mTOR, and the development of NAFLD.

Metformin has been used as a hypoglycaemic agent to treat diabetes mellitus and NAFLD remission affected by IR, but studies on this agent in relation to lipid metabolism are not common. Recently, metformin was found to negatively regulate mTORC1 through AMP-activated protein kinase (AMPK), resulting in the expression and dissociation of SREBP-1c and fatty acid oxidative damage and the reduction of liver lipid content [39]. It is speculated that an AMPK activator (metformin) may affect the expression of lipogenic genes through an mTORC1—SREBP-1c-dependent mechanism, thereby reducing fat deposition in the liver. Bentzinger et al. [40] demonstrated that rapamycin treatment inhibited the nuclear accumulation of SREBP-1c and the expression of lipogenic SREBP-1c target genes in cells and mice. These results confirm the finding that mTOR deletion is associated with the reduced expression of the lipid-forming gene SREBP-1c [35,41]. In addition, the nuclear receptor peroxisome proliferator-activated receptor γ (PPARγ), which controls lipid homeostasis, is impeded by mTORC1 inhibition [42]. However, whether other genes in the PPAR family are also associated with mTOR in coordinating the regulation of NAFLD remains unclear. Future research on lipid metabolism in NAFLD is necessary.

Pancreatic progenitor cell differentiation and proliferation factors regulate mTORC1 signaling in lipid metabolism; specifically, these processes and factors inhibit mTORC1 activity by blocking the decrease in Raptor ubiquitination, thereby inhibiting hepatic steatosis [43]. In this previous study, the relationship among Raptor, mTOR structure, and NAFLD-related lipid metabolism was not explored at the level of the overall mTORC1 structure. Further research on the relationship between NAFLD and the structure of mTORC components, such as mLST8, mSIN1, Deptor, Kinase, and Rictor, is warranted (Figure 2).

## 4. mTOR Regulates Hepatic IR through Foxo1 and Lipin1

Insulin resistance (IR) is a major factor in the progression of NAFLD. IR in the liver seriously affects energy balance and metabolism, resulting in excessive lipid accumulation in the liver, which is the main cause of fatty liver. In the underlying mechanism, insulin-mediated hepatic glucose metabolism reduces hepatic gluconeogenesis by directly regulating the activation of phosphatidylinositol-dependent kinase-1 (PDK1) and mTORC2, resulting in the phosphorylation of protein kinase B (Akt), which activates hepatic cell insulin receptors. Phosphorylation of Foxo1 triggers liver glycogen synthesis and downregulates the transcription of glycogen-producing enzymes. Moreover, mTORC1 can regulate Foxo1 phosphorylation and transcription. Therefore, we proposed that IR is regulated in NAFLD by a closed-loop pathway, with mTOR at the center and Foxo1 as the connection involved in the joint regulation of IR.

Rajan et al. [44] concluded that feedback regulation of insulin receptor substrate 1 (IRS1) by mTORC1 and between IRS1 and insulin are the key factors underlying the occurrence of IR. This conclusion suggests that mTOR plays a key role in the development of IR. In addition, through whole-system experiments and the model analysis of Foxo1-related insulin signal transmission, Rajan et al., verified that the attenuation of mTORC1-to-IRS1 feedback is the main mechanism of IR in the diabetic state. Specifically, numerous islet beta cells are required to ensure normal blood glucose in NAFLD patients, which causes the high fasting insulin concentration in these patients. The relationship between islet beta cells and type 2 diabetes mellitus has been widely studied, but such a relationship has not been found in NAFLD; this topic may warrant future research efforts to block the development of IR in NAFLD.

The mTORC1 complex subunit, Raptor, has been shown to specifically block the lipid origin of insulin Akt signaling at pleckstrin homology domain leucine-rich repeat-containing protein phosphatase 2 (PLHPP2), which targets Akt at Ser473, thereby blocking the progression of NAFLD. Han et al. [45] identified an mTOR-mediated pathway of nutritional reactivity that links insulin signaling to the upregulation of the hepatic lipid metabolism SREBPs gene family. This relationship is mediated by cyclic AMP response element-binding protein (CREB) transcription coactivator 2 (CRTC2). Other data suggest that CRTC2 and Raptor [45,46] are potential new targets for regulating glucose-mediated insulin secretion. Lipin1 [47], a key regulator of lipid metabolism that is located downstream of mTOR, is not only a phospholipid acid phosphatase necessary for triglyceride biosynthesis but also a transcriptional coactivator that can regulate the expression of genes related to lipid metabolism; in addition, Lipin1 was found to be closely related to the development of diabetes and IR. A previous study [47] showed that enhancing mTORC2 activity in liver cells increases Lipin1 expression and correspondingly escalates IR while inhibiting CREB, with A-CREB decreasing Lipin1 expression. These findings suggest that the mTORC2/CREB complex coregulates Lipin1 expression in liver cells, thereby affecting lipid accumulation in NAFLD. In addition, when Lipin1 expression is elevated in healthy humans, protein kinase C epsilon (PKCε) mediates intracellular diacylglycerol (DAG) accumulation to inhibit the insulin signaling pathway, which reduces liver glucose production and further inhibits IR. This process also somewhat hinders the development of NAFLD.

Insulin has both direct and indirect effects on the liver; in particular, insulin indirectly reduces the transport of fatty acid to the liver by inhibiting the lipolysis of white fat and reducing the hepatic content of acetyl-CoA, an allosteric activator of pyruvate carboxylase, thus affecting the progression of NAFLD. However, although it is known that the mTORC2–Akt signal transduction to ATP-citrate lyase drives the formation of brown fat from DNL [48], there is a large gap in the knowledge regarding white fat and mTOR (Figure 2).

## 5. mTOR Regulates OS through PIG3, P53, JNK

Liver diseases, including NAFLD and fatty liver disease (FLD), are closely related to oxidative stress (OS). Researchers found that the increased delivery of free fatty acid (FFA) to the liver triggered an increase in mitochondrial OS [49]. OS can promote liver damage and inflammation in NAFLD, and the vicious cycle between NAFLD and OS was demonstrated using the antioxidant vitamin E [50]. In summary, excessive OS in the body will aggravate NAFLD and even NASH and will further induce OS after NAFLD has developed.

OS can be assessed directly by ROS assays or indirectly by the analysis of the associated lipid, protein, and nucleic acid damage caused by excessive ROS production. Researchers [51] treated liver cells in vitro with hydrogen peroxide (H_2_O_2_) and found increased ROS levels and the phosphorylation of nuclear factor erythroid 2-related factor 2 (Nrf2), JNK, and p53. To investigate the relationship among ROS, Nrf2, JNK, and p53, we exposed H_2_O_2_-treated cells to the JNK inhibitor SP600125 and the antioxidant N-acetylcysteine and found that both treatments inhibited the H_2_O_2_-induced increases in JNK and p53 phosphorylation and cell death. Therefore, we hypothesized that H_2_O_2_ may regulate hepatocyte death through the OS-JNK-p53 pathway. The discovery of the bioactive substance, quasi-polar acid B, extracted from hawthorn root bark, confirmed this hypothesis; the extract mainly affected autophagy by inhibiting the OS-JNK-p53 positive feedback loop triggered by mTOR [52], thus reducing the occurrence of NAFLD and promoting HCC apoptosis. Other researchers [53] elaborated on the interaction between OS and p53 and verified the above conjecture. Other studies have shown that the induction of p53 expression may depend on the activation of the PI3K/Akt/mTOR pathway and OS, which regulates hepatocyte apoptosis [54].

PIG3 is closely related to NADPH quinone oxidoreductase, which induces OS [55] and reduces the mitochondrial membrane potential. Silencing PIG3 was proposed to increase the expression of phosphatase and the tensin homologue (PTEN) and reduce the phosphorylation of phosphatidylinositol 3-kinase (PI3K) and Akt, suggesting that PIG3 induces OS at an intermediate level and affects Akt phosphorylation and mTOR activation [56]. The PIG3-p53-OS axis was proposed to be a regulatory loop that connects to the OS-JNK-p53 positive feedback loop described above to form the OS-JNK-p53-PIG3 pathway; this mechanism is regulated by mTOR, thus affecting NAFLD progression. One study found that PTEN can convert phosphatidylinositol 3,4,5-triphosphate (PIP3) to phosphatidylinositol 4,5-diphosphate (PIP2), negatively regulate the PI3K/Akt signaling pathway, and downregulate or terminate insulin signaling downstream of PI3K [57], which is involved in the development of NAFLD. Moreover, PTEN can directly affect cellular ROS levels; thus, increasing PI3K expression indirectly affects mTOR activity [58,59]. Therefore, the PI3K/AKT/PTEN pathway leads to OS and the development of NAFLD by promoting oxidatively-stressed liver cells to undergo apoptosis, which is considered to play a key role in the pathogenesis of NAFLD [60,61]. In fact, the association between the loss of liver-specific PTEN expression and NAFLD/liver fibrosis was confirmed in 2005 [62]. Moreover, the role of PTEN in the development of NAFLD and NASH was found to be based on the PI3K/AKT pathway [63]. The activation of mTOR can affect the expression of nuclear factor-κB (NF-κB) and further inhibit PTEN gene expression [64]. The above results further support the hypothesis that mTOR can affect NAFLD by regulating the OS-JNK-p53-PIG3 pathway.

In addition, Dabin Liu [65] proved experimentally that recombinant squalene monooxygenase (SQLE) can induce the silencing of PTEN, the upstream regulatory gene in the PI3K/Akt pathway, leading to the activation of the PI3K/Akt/mTOR pathway to promote NAFLD-associated HCC. Chronic endoplasmic reticulum stress, associated with OS, also drives the progression of NAFLD. Forkhead box A3 (FOXA3) is a member of the liver nuclear transcription factor family that plays a key role in metabolic homeostasis. The loss of FOXA3 function in mice has been shown to inhibit diet-induced chronic endoplasmic reticulum stress, fatty liver, and IR. Mechanistic analysis showed that FOXA3 promoted the expression of adipogenic genes, including SREBP-1c, and enhanced lipid synthesis. Elevated levels of FOXA3 and SREBP-1c in the liver were found in both obese mice and NAFLD patients. This study confirmed that FOXA3 links stress to NAFLD progression, demonstrating the importance of the FOXA3/SREBP-1c transcriptional axis in NAFLD progression, proposing that FOXA3 is a potential therapeutic target for FLD [66]. However, there have been few studies on the relationship between mTOR and endoplasmic reticulum stress in NAFLD. At present, it is unclear whether this lack of knowledge relates to the challenges of identifying a correlation or to the bridge between mTOR and endoplasmic reticulum stress; further study is required (Figure 2).

## 6. mTOR Regulates the Intestinal Flora through TLRs

Recent studies have indicated that intestinal floras play an important role in the pathogenesis and progression of NAFLD [67], and the mechanism may be related to improving energy metabolism, increasing IR, and regulating bile acid and choline metabolism. The therapeutic effect of intestinal probiotics on NAFLD has been repeatedly confirmed [68] to mainly involve intestinal dysbiosis. When the intestinal barrier is damaged, bacterial endotoxin lipopolysaccharide (LPS) can pass through the portal vein into the blood and then into the liver, wherein it can cause sepsis. LPS is recognized and bound by TLRs (e.g., TLR4 and TLR9) on the surface of hepatocytes or Kupffer cells [69], which subsequently activate an inflammatory cascade that ultimately leads to hepatic steatosis and NASH progression [70]. In addition to the intestinal flora, intestinal metabolites such as short-chain fatty acids [71] have also been reported to be involved in the pathogenesis of NAFLD.

Upon activation, macrophages can release miR-192-5p–enriched hepatocellular exosomes that inhibit the expression of Rictor and the phosphorylation of Akt and FoxO1. Macrophage polarization towards a pro-inflammatory phenotype (M1) represents an important event in the progression of NAFLD [72]. Seif El-Din et al., found that the probiotic and metformin treatment of early-stage NAFLD can target intestinal microbiome dysregulation and the p-Akt/mTOR/LC-3II pathway to reduce liver damage in NAFLD rats. Other researchers [73] reported that the highly enriched Escherichia coli strain, NF73-1, isolated and cultured from NASH patients, not only accelerated the progression of NAFLD but also translocated it to the liver, resulting in the increased production of liver M1 macrophages through the TLR2/NLRP3 pathway. The subsequent activation of mTOR-S6K1-SREBP-1c/PPARα signaling led to metabolic conversions of triglyceride oxidation to triglyceride synthesis in NAFLD mice. Intestinal bacterial depletion does not only affect the liver; 5-hydroxytryptamine (5-HT) in the brain can also regulate intestinal movement, permeability, and other functions and is considered an important central physiological modulator of intestinal function. Dysfunction of the intestinal serotonergic system leads to the impairment of the intestinal barrier, which promotes further bacterial endotoxin (LPS) translocation into the liver, thereby contributing to the development of NAFLD. Tryptophan, a precursor of 5-HT, also plays an important role in intestinal homeostasis and energy metabolism. A previous study [74] showed that tryptophan supplementation can enhance 5-HT production, thus aggravating fatty liver degeneration, and that 5-HT can activate the mTOR signaling pathway in mice fed on a high-fat, high-fructose diet. Therefore, mTOR can regulate the intestinal barrier damage caused by 5-HT, allowing LPS leakage into the blood and thus alleviating NAFLD inflammation. In addition to the intestinal flora, the gut hormone ghrelin has been found [75] to restore increased autophagy and inhibit NF-κB nuclear translocation through AMPK/mTOR, thereby slowing the progression of NAFLD.

Scientists at the Sahlgrenska Institute of the University of Gothenburg, Sweden [76], have shown that histidine-derived imidazole propionate in the gut drives insulin signaling at the level of insulin receptor substrates by activating p38 mitogen-activated protein kinase (MAPK), thereby promoting the phosphorylation of p62 and the subsequent activation of mTORC1. These findings also confirm that the microbiological metabolite imidazole propionate may contribute to the pathogenesis of type 2 diabetes and may even affect NAFLD and other metabolic diseases. At present, there is considerable interest in researching the intestinal flora in NAFLD, obesity, type 2 diabetes, and other metabolic diseases, although the intestinal flora is complex and readily changeable in response to alterations in the external environment. Research on this topic is indeed lacking. Therefore, this manuscript does not provide a comprehensive summary, but the data confirm the importance of the influence of mTOR, whether direct or indirect, on the intestinal flora in the enterohepatic axis in the context of NAFLD (Figure 3).

## 7. mTOR Regulates Liver Autophagy

Autophagy, or impaired autophagy, is associated with NAFLD, but the underlying mechanism of autophagy dysfunction in NAFLD is not completely clear [77]. The regulation of autophagy is very complex, involving many signaling pathways and mediators, such as programmed cell death-1, Beclin-1, UV radiation resistance-associated gene (UVRAG), mTOR, and p53 [78]; all of these autophagy factors are closely related to the development of NAFLD.

mTOR, an important signal sensor in cells, also plays an important role in autophagy [79]. When mTORC2 expression was inhibited in mice, activated Akt increased mTORC1 activity and inhibited autophagy by affecting TSC2–Rheb [80,81]. It has also been reported that mTORC2 inhibits Foxo1/3 activity by reducing Akt activity, thereby reducing the production of autophagy proteins, inhibiting the induction of autophagy and inhibiting the development of liver tumors [82,83]. mTORC1 and mTORC2 seem to play equally important roles in autophagy. Thus, it was hypothesized that mTOR activation negatively correlates with autophagy.

Zhang Y et al. [84] recently reported that the PI3K/mTOR pathway is an important participant in the autophagic regulation of lipid metabolism. Bhalla et al. [85] reported that a panhistone deacetylase inhibitor, PCI-24781, induced autophagy in diffuse large B-cell lymphoma (DLBCL) cells through hypoxia-inducible factor 1α (HIF-1α) and that this effect was mediated by the PI3K/AKT/mTOR pathway. Singh R et al. [86] found that under normal physiological conditions, mTOR inhibits autophagy to prevent the breakdown of newly synthesized cellular components. To accomplish this, mTORC1 inhibits the phosphorylation of unc-51-like autophagy-activated kinase 1 (ULK1) and autophagy-related protein (ATG) 13, which are two key markers of the early induction of autophagy [87,88,89]. The above results confirm our “negative correlation” conjecture described above. When an organism is receiving adequate nutrition, mTORC1 can downregulate the UVRAG gene through exosomal miR-10A-5p from bone marrow mesenchymal stem cells, thus blocking the initiation of autophagy. This phosphorylation level can be used as a marker to detect early and late autophagy [90].

In 2009, a research group led by Yoshimori [91] at Osaka University found that Rubicon is related to the inhibition of autophagy. Rubicon knockout (KO) in mice eliminated autophagy-related liver disease. Moreover, when Rubicon protein expression was inhibited, model animals showed elevated levels of autophagy. In studying the mechanism of this phenomenon, Tanaka S et al. [92] confirmed, in 2016, that Rubicon overexpression led to the accelerated apoptosis of liver cells, excessive accumulation of lipids and inhibition of autophagy, thus suggesting that this protein plays a coordinated pathogenic role in NAFLD. Moreover, in NAFLD mice, the addition of the calcium-dependent protease calpain-2 resulted in the degradation of recombinant ATG7 in the key ATG12-ATG5 complex involved in the phagocytic membrane extension of autophagy vesicles, resulting in defects and reductions in autophagy and diminished NAFLD progression [93]. However, more evidence is needed to determine whether this process is modulated by mTOR.

It has been widely documented that the induction of autophagy can prevent the palmitate-induced death of hepatocytes [94,95,96]. As mTORC1 is a major physiological inhibitor of autophagy, the mTORC1 inhibitor, rapamycin, can stimulate autophagy in liver cells and liver [97]; thus, it is reasonable to speculate that the protective effect of inhibiting lipid toxicity may be due to the induction of autophagy by mTORC1. Zhang S et al. [98] reported that AMPK/mTOR-mediated autophagy was significantly inhibited in high-fat diet-fed mice and FFA-treated LO2 cells, indicating that AMPK/mTOR mediates autophagy levels and thus reduces hepatotoxicity. These results suggest that the mTOR-mediated activation of autophagy is an important protective mechanism against NAFLD. In addition, liraglutide was reported to increase the number of autophagosomes in the body, and this process is believed to be mediated by the AMPK/mTOR signaling pathway [99]. Further studies showed that mice with fibronectin type III domain-containing protein 5 (Fndc5) gene deletion had increased liver fat deposition, fatty acid oxidation, and decreased autophagy, while the lentiviral overexpression of Fndc5 significantly increased fatty acid oxidation and autophagy and alleviated liver steatosis in obese mice. The mechanism by which hepatic steatosis was reduced was found to involve decreased mTORC1 activity and consequently increased AMPK phosphorylation, which promoted the expression of fatty acid oxidation-related target genes [100].

In conclusion, autophagy is closely related to NAFLD, so we expect mTOR to be a target for autophagy intervention, and further research will hopefully provide important clues for the prevention and treatment of liver diseases (Figure 3).

## 8. mTOR Regulates Liver Inflammation

During the progression of NAFLD, systemic inflammation occurs, especially in the development of obesity-related NASH; the inflammatory response leads to the recruitment of adhesion factors, cytokines, and chemokines through the activation of signaling pathways. Inflammatory factors such as tumor necrosis factor-α (TNF-α), IL-1β, IL-1, IL-12, IL-18, and IL-6, attack the liver [101,102]. Therefore, inflammatory stress is not an independent risk factor for the development of NAFLD.

To test this hypothesis, LPS-induced macrophage models of liver autophagy and NASH were treated with scoparone, which suppressed the ROS/p38/Nrf2 axis and the PI3K/AKT/mTOR pathway and enhanced autophagic flux by coregulating autophagy and inhibiting inflammation [103]. In addition, Yes-associated protein (YAP) [104], which is closely associated with mTOR, was reported to play a key role in the progression of NASH through the production of pro-inflammatory cytokines and the persistence of liver inflammation, providing insights into the mechanism of LPS-mediated inflammation in NAFLD progression through the LPS/TLR4 signaling pathway [105]. This mechanism provides further insight into the link between intestinal microbiota (LPS) and the Hippo pathway, suggesting roles for the Hippo pathway outside of cancer and for LPS beyond the classical enterohepatic axis.

Studies have shown that inflammation exacerbates the progression of NAFLD by disrupting cholesterol homeostasis. To test this hypothesis, researchers [106] induced chronic inflammation by subcutaneously injecting 10% casein into apolipoprotein E (ApoE) KO mice or stimulating HepG2 hepatoblastoma cell lines with IL-1β. The results showed that inflammation increased lipid accumulation in HepG2 cells and in the liver of ApoE KO mice. Further analysis showed that inflammation increased the phosphorylation of mTOR, eukaryotic initiation factor 4E (eIF4E)-binding protein 1 (4E-BP1), and p70 S6 kinase (p70S6K). However, the specific mechanism involving mTOR and inflammation in NAFLD remained unclear. Therefore, researchers [107] stimulated human hepatoblastoma HepG2 cells with TNF-α and IL-6 in vitro. In C57BL/6J mice injected with casein, inflammatory stress enhanced the phosphorylation of mTOR and its downstream translational regulators (including p70S6K, eIF4E, and 4E-BP1). However, rapamycin, an mTOR specific inhibitor, reduced the phosphorylation of mTOR signaling pathway components and the translation efficiency of CD36 and its protein levels, even under inflammatory stress, thus reducing liver lipid accumulation caused by inflammatory stress. In addition to mouse and cell experiments, Sapp V. et al. [108] treated zebrafish larvae with fructose and found that fructose-treated fish showed the activation of inflammatory and adipogenic genes. The subsequent treatment of zebrafish with rapamycin prevented the development of hepatic steatosis, as did chlamycin and valamycin. Another experiment showed that inhibition of ghrelin o-acyltransferase could block the inflammatory response by regulating the mTOR pathway and thus reduce hepatic cell steatosis, which confirmed that mTOR can affect NAFLD by regulating the inflammatory response [98] (Figure 3).

## 9. mTOR Regulates Liver Genetic Polymorphisms and Epigenetics

Genetic polymorphisms and the epigenetic background are also considered key factors affecting the occurrence and progression of NAFLD, and approximately half of variation in liver fat content can be explained by genetic factors [109]. Although the specific pathogenesis is currently unclear, some scholars have proposed that genetic factors can explain the pathogenesis of lean NAFLD [110]. Patients with lean NAFLD were found to have the same typical pathological changes as those with obese and overweight NAFLD, but these pathological changes were less severe, and the prognoses were better in lean NAFLD than obese NAFLD [111]. The results of this study confirm the influence of genetics on the prognosis of lean, obese, and overweight NAFLD.

Liu D et al., performed RNA sequencing analysis of NAFL-associated HCC samples and found that SQLE was the most overexpressed metabolic gene in these patients [65]. Moreover, mouse hepatocyte-specific SQLE transgenic expression accelerated the development of HCC induced by a high-fat, high-cholesterol diet. Importantly, SQLE upregulation is involved in OS-induced DNA methyltransferase 3A (DNMT3A) expression, DNMT3A-mediated epigenetic silencing of PTEN, and Akt–mTOR activation, which together triggers NAFLD.

Odd skipped-related 1 (Osr1) is a novel tumor suppressor gene identified in several cancer cell lines [112]. A previous study found that [113] Osr1^+/−^ mice had more severe FLD, with more severe steatosis and hepatocellular balloon degeneration and increased macrophage infiltration. In addition, Osr1+/− mice showed increased TGFB and Fn1 expression and collagen fiber deposition between the central and portal vein bundles, indicating progression towards NASH. These results suggest that Osr1 inhibits the development of NAFLD by regulating liver inflammation through the PI3K/AKT/mTOR, NF-κB, and JNK signaling pathways.

De Conti A et al. [114] reported that extensive transcriptome and epigenetic changes in protein-coding genes during the development of NASH-derived HCC are potentially important diagnostic and therapeutic targets for HCC. To confirm this conclusion, we obtained liver samples from C57BL/6J mouse models of NASH-associated HCC to explore the role of microRNA (miRNA) in the pathogenesis of this disease. The altered miRNAs were mainly related to the activation of the extracellular signal-regulated protein kinase 1/2 (ERK1/2), mTOR, and epidermal growth factor (EGF) signaling pathways in HCC. However, it remains unknown whether changes in miRNA levels that regulate one of these three pathway or that jointly regulate all three affect the occurrence of liver cancer; further research is needed.

Santana et al. [115] characterized the mutation spectrum of HCC and found extensive mutations in the Wnt/β-catenin, MAPK, and PI3K–Akt–mTOR pathways. Genetic alterations were associated with 47.5% of HCC cases, and these alterations included 190 single nucleotide variants and 5 insertions/deletions. The Wnt/β-catenin pathway was affected by 30% of the identified mutations, followed by the MAPK (27.5%) and PI3K-Akt-mTOR pathways (25%). FAT1 and APC were the most commonly mutated genes (17.5%), followed by RAS mutations (12.5%), but no association between the BRAF mutation spectrum and clinicopathological features was found. The mutation spectrum of HCC is not comprehensive as the sequenced specimens were from surgical HCC resections performed from 1998 to 2017; these data need to be updated and further explored.

Epidermal growth factor receptor (EGFR) promotes cell proliferation through the PI3K-Akt-mTOR signaling pathway and participates in the occurrence and development of HCC. Zhang L et al. [116] confirmed that single-nucleotide polymorphisms (SNPs) in the EGFR 3ʹ-UTR are associated with HCC risk. Compared with the TT genotype, the TG and GG genotypes of SNP RS884225 in EGFR significantly reduce the risk of HCC and are associated with decreased tumor size. An increased number of G alleles is also associated with reduced HCC risk. In addition, EGFR mRNA expression in HCC patients has been found to be lower in patients with the TG or GG genotype. Bioinformatics analysis has shown that mutant EGFR binding to miR-3196 is weaker in patients with the TT genotype than in those with the TG or GG genotype. In summary, mTOR has less direct means of regulating genetic polymorphisms and epigenetics, and most genetic and epigenetic changes regulate the development of NAFLD through the coregulation of multiple factors. However, mTOR does have a certain correlation with genetic polymorphisms and epigenetics, but additional research is needed (Figure 3).

## 10. Crosstalk between mTOR and Novel Pathways: A Source of Potential New Targets for the Treatment of NAFLD-Associated HCC

In hepatocytes, Notch was found to be a contributing factor to NASH-associated fibrosis, and its role in HCC was confirmed in later research [117]. Notch signaling is involved in mTORC2 expression, proliferation, differentiation, and development [118]. Notch signaling has previously been shown to activate the mTOR pathway in the haematopoietic system [119], and the inhibition of mTOR can upregulate Notch [120,121], suggesting crosstalk between Notch and mTOR signaling in other tissues. However, how these interactions translate into the initiation and promotion of liver cancer still needs to be clarified. Later research revealed that Notch signaling and mTOR exhibit crosstalk in fatty liver. An acute reduction in hepatic Notch signaling increases insulin sensitivity and decreases mTORC1-mediated SREBP-1c activity and hepatic triglycerides, and the inhibition of mTOR prevents Notch-induced FLD [122]. Previous studies have found that Notch activation in the stomachs of mice may be upstream of mTOR in the Notch-mTOR signaling cascade. Activation of the Notch and mTOR pathways was observed in both mouse and human stomach cancer, suggesting that these pathways may be involved in tumorigenesis because they both play an important role in regulating cell proliferation. In addition, this previous study found that the inhibition of Notch or mTORC1 alone in human gastric cancer cell lines resulted in reduced cell growth, while combined pathway inhibition further inhibited cancer cell growth, suggesting that Notch and mTOR synergistically promote gastric cancer cell proliferation and that this sustained proliferation is cancer dependent on both pathways [123].

The abnormal activation of Hedgehog signaling has been shown to promote HCC cell proliferation, migration, and invasion, suggesting that this pathway plays a key role in the progression of HCC [124,125,126]. However, a recent study in November 2021 found that the Hedgehog signaling pathway is involved in lipid metabolism through the direct regulation of the previously unidentified, long noncoding RNA, Hilnc, which plays an important role in lipid metabolism [127]. Liver lipid metabolism is one influential factor in the “two strike” theory of NAFLD pathogenesis, and mTOR plays a decisive role in many aspects of lipid metabolism. Therefore, these researchers hypothesized that crosstalk between mTOR and the Hedgehog signaling pathway can regulate NAFLD and even HCC through lipid metabolism. The relationship between mTOR and the Hedgehog pathway in esophageal cancer was described in 2013 [128]. This previous study found that both AKT and S6K1, which are upstream and downstream of the mTOR pathway, respectively, exhibit crosstalk with the Hedgehog signaling pathway to activate glioma-associated oncogene homologue 1 (Gli1) through mTOR regulation. In this study, Gli1 was identified as a substrate of S6K1, thus providing a mechanism by which the mTOR and Hedgehog signaling pathways activate Gli1 in a manner not dependent on Smoothened, a G protein-coupled receptor signal transducer. Surprisingly, the combination of the mTOR pathway inhibitor RAD-001 and the Hedgehog pathway inhibitor GFC-0449 inhibited esophageal cancer cells by up to 90% [129].

Other studies have found that the Hippo pathway is critical for inducing apoptosis and inhibiting proliferation and animal development [130]. The Hippo pathway inhibits proliferation and activates apoptosis by inhibiting the downstream effector YAP. Since cell growth and proliferation must be closely coordinated, crosstalk between the Hippo and mTOR pathways has been reported, but it was not until 2012 that Tumaneng K et al. [131] reported that YAP positively regulates both mTORC1 and mTORC2. The role of the transcriptional coactivator with the PDZ-binding motif (TAZ), a key activator of the Hippo pathway, in HCC also supports this crosstalk. TAZ KO in HCC cell lines inactivates the mTOR pathway and slows growth [132]. The first paper on the relationship between the Hippo pathway and autophagy was published in 2014; this publication reported that mTORC1 signaling was the main upstream inhibitory pathway of autophagy and remained in a constitutively active state in TSC1-deficient cells. The researchers found that autophagy and the Hippo pathway were significantly inhibited in TSC1-deficient cells. Further mechanistic studies showed that YAP could interact with the autophagy receptor p62, which promoted the lysosomal degradation of YAP. In other words, the autophagy pathway leads to Hippo pathway activation by degrading YAP [133]. Then, in 2019, researchers discovered the mechanism involving mTOR, autophagy, and the Hippo pathway, in HCC. YAP can inhibit the initiation of autophagy by activating the mTORC1 signaling pathway, which proves that the mTORC1 signaling pathway can both regulate the Hippo pathway and be regulated by the Hippo pathway. Elucidation of the relationship between mTOR and the Hippo pathway and the origin of autophagy confirmed our conjecture that mTORC1 and Hippo pathway crosstalk can inhibit HCC growth and promote HCC cell apoptosis through the autophagy pathway [134]. It is not known whether this mechanism is related to NAFLD or HCC, although researchers reported in 2020 that Raptor can be directly phosphorylated by the core kinase of the Hippo signaling pathway after receipt of a growth inhibition signal, thereby weakening mTORC1 activity and inhibiting cell and organ growth [135]. Although the relationship between the Hippo and mTOCR1 signaling pathways has not yet been conclusively proven, crosstalk between these pathways represents a new entity to study regarding the regulation of organ development. Such research may provide feasible strategies for liver regeneration and hepatectomy in late-stage NAFLD.

In conclusion, the Hippo, Hedgehog, Notch, and mTOR pathways are closely related to cancer. Although there are few studies on the occurrence and development of NAFLD-associated HCC, these pathways may provide new therapeutic options and research directions for the clinical treatment of this disease. In addition, research in esophageal cancer and gastric cancer, and findings regarding crosstalk between mTOR and both Notch and Hedgehog, have confirmed the related hypotheses. Crosstalk between Hippo, Hedgehog, Notch, and mTOR is closely related to the occurrence and development of liver cancer. Moreover, dual inhibitor combination therapy for liver cancer and other types of cancer is eagerly anticipated. However, as there have been few studies on the relationships of mTOR with Hippo, Hedgehog, and Notch, the summary presented herein is not comprehensive. As research progresses, future reviews will provide updates. How these pathways interact to dysregulate cell proliferation to support tumorigenesis, and the potential therapeutic targeting of these pathways in combination, remain important areas of future research (Figure 4).

## 11. Discussion

mTOR has received much attention since its discovery. Because mTOR is closely related to biological activities, not only in the liver as described in this paper but also in fat, skeletal muscle, brain, gastrointestinal and other organs, mTOR-targeted inhibitors have been studied. However, at present, the main therapeutic application of mTOR is in cancer, especially focused on the classic PI3K/AKT/mTOR signaling pathway, implicated in cancer, which has been widely studied to develop treatments for cancer. However, the focus on the development of cancer therapeutics has come at the cost of adequate research on metabolic diseases and other pathologies. There is currently a lack of systematic elucidation of the relationship between mTOR and metabolic diseases, such as NAFLD; thus, this review aims to provide an update on research on NAFLD in metabolic diseases and advocates for a shift in research attention to address the crucial need for more research and development of mTOR inhibitors for application in metabolic diseases.

This review can not only solve the early noninvasive diagnostic indicators and clinical future targets so as to provide a certain reference basis, but also sheds light on the molecular mechanism by means of macro and micro clear concrete mechanisms and the effect of the mTOR NAFLD, summed up in the form of a table (Table 1), and mTOR signaling pathways/mechanism of NAFLD stages and the influence of the development process. The effect of crosstalk between mTOR and novel pathways on the treatment of non-alcoholic fatty liver-related hepatocellular carcinoma is also of great significance for the treatment of NAFLD-related hepatocellular carcinoma in the future, which may provide a reference for the development of new mTOR inhibitors. According to the table (Table 1) and the above explanation, we can also find that there are many factors affecting the development of NAFLD, but most of them focus on the signaling pathways such as PI3K/AKT/mTOR, AMPK/mTORC1, and mTORC1/SREBP1c/DNL, which directly or indirectly affect the lipid accumulation and mediate the level of autophagy in the liver. Therefore, this paper believes that the developmental process of NAFLD is closely related to liver lipid metabolism, regardless of the “second hit” or “multiple hit” theory of NAFLD, with mTOR regulating liver lipid metabolism through its upstream, downstream, and other ways. Therefore, mTOR is of great significance for future drug development and mechanism research regarding NAFLD.

However, based on the influence of mTOR on gut microbiota, genetic polymorphisms, and epigenetics on the development of NAFLD, this paper has not fully elaborated on this area which still needs researchers to continue their efforts. As there are few studies on these two aspects, this indicates that when there is a bottleneck in the studies of lipid metabolism, the effects of mTOR-based regulation of gut microbiota, genetic polymorphisms, and epigenetics on NAFLD can be considered from another perspective. At present, there are many studies on the crosstalk between mTOR and novel pathways in cancers such as gastric cancer and esophageal cancer, while there are few studies on the influence of hepatocellular carcinoma. Due to the limitations of such conditions, this paper is not comprehensive enough. With more in-depth research, we will further follow up and review this area. In addition, whether the crosstalk between mTOR and novel pathways also has a certain impact on NAFLD and NASH remains to be discovered in further studies.

Decades of research on molecular mechanisms, especially the recent discovery and development of transcriptomics, metabolomics, proteomics, and other technologies, have greatly deepened our collective understanding of NAFLD. However, challenges remain regarding the noninvasive diagnosis of NAFLD and the lack of specific drugs; thus, this review will provide a reference for identifying early noninvasive diagnostic indicators and clinical targets in the future:(1)Since mTOR is closely related to various influencing factors in NAFLD, it is speculated that mTOR can be used as one of the early noninvasive diagnostic indicators.(2)The molecular mechanism of mTOR regulating NAFLD was elaborated upon.(3)The therapeutic effect of crosstalk between mTOR and Hippo, Hedgehog, and Notch on NAFLD-HCC was predicted.(4)mTOR can be used as a new therapeutic target for NAFLD in clinic.

## 12. Conclusions

In this review, we present and discuss evidence for the role of mTOR in the progression of NAFLD; data indicate that mTOR regulates NAFLD progression through lipid metabolism, IR, OS, intestinal flora, autophagy, the inflammatory response, genetic polymorphisms, and epigenetics. The role of crosstalk between mTOR and the upstream regulators, Notch, Hedgehog, and Hippo, related to the occurrence and development of NAFLD-associated HCC and the possibility of mTOR-specific therapeutic strategies, are elaborated upon in detail within. In addition, the specific influence of mTOR on NAFLD is hypothesized to involve both micro-regulation (the mechanism by which mTOR regulates factors that influence NAFLD) and macro-regulation (the relationship between influencing factors), which together summarize the influence of mTOR on the development of NAFLD. The evidence proves the importance of mTOR in NAFLD.

From the micro-regulation point of view, mTOR regulates the mechanisms that influence NAFLD. Regarding lipid metabolism, mTOR blocks not only FFA formation and DNL, the source of abnormal liver lipid metabolism, but also the transcription of INSIG, the key gene determining SREBP production, thus inhibiting the development of NAFLD by altering the metabolism. From the perspective of IR, mTOR regulates IRS1, Lipin1, Foxo1, PKCε, and other important targets related to IR and plays a key role in the development of IR by affecting insulin production, thus blocking the progression of NAFLD. Regarding OS, mTOR plays a decisive role in the regulation of mitochondrial β oxidation, as well as of PTEN and other related targets, and this review summarizes evidence that the PIG3/p53/JNK/OS pathway plays an important role in NAFLD. Another related topic that requires further study is the direct regulation of mTOR by intestinal microbiota in the enterohepatic axis. mTOR can affect lipid metabolism and autophagy in the liver by regulating intestinal microbiota-related receptors, such as those in the LPS/TLR classic pathway and is closely related to anti-inflammatory responses. Regarding autophagy, the relationship between mTOR and autophagy was first discovered in cancer, and studies have found that mTOR activation is negatively correlated with autophagy. Further study confirmed this result in the development of NAFLD. mTOR directly controls the development of multiple autophagy-related complexes, such as the ULK1 and VPS34 complexes, and the levels of important autophagy-related proteins, such as TXNIP, AMPK, UVRAG, and p53, to regulate autophagy, thus affecting the development of NAFLD and HCC. Inflammatory responses are evident throughout the development of NAFLD; in the early stages of NAFLD, lipid accumulation in the cytoplasm of liver cells (the first attack) triggers a series of cytotoxic events (the second attack), leading to an inflammatory response in the liver, and such responses are also observed in the later stages of disease development. The induction of NASH by an excessive inflammatory response is closely related to the nature of the inflammatory response. We found that mTOR can directly regulate various inflammatory factors, such as IL-6, IL-12, IL-1, and other members of the IL family, TNF-α, NF-κB, and JNK, which directly affect the pathogenesis of NAFLD by blocking disease development. Finally, data show that liver genetic polymorphisms and epigenetics are directly regulated by mTOR through the WNK and MAPK pathways and are indirectly regulated by this protein through the regulation of miRNA and Osr1, thus affecting the development of NAFLD. However, the mechanism by which genetic polymorphisms and epigenetics impact the occurrence and development of NAFLD has not been fully elucidated, and in-depth mechanistic studies will hopefully identify new treatment strategies to benefit NAFLD patients. We also find that this is not independent of the above influencing factors, such as the expression of the DNMT3A gene, with the DNMT3A-mediated PTEN gene having importance as a target protein of mTOR in regulating oxidative stress. Another important gene that affects genetic polymorphisms, Osr1, is involved in the regulation of liver inflammation through the PI3K/AKT/mTOR, NF-κB, and JNK signaling pathways and is closely related to liver inflammation.

From the macro-mediators, which each influence various factors, the relationship between the intestinal flora, via enterohepatic shaft TLR family receptors, can influence the autophagy and inflammation of the liver, and further affect the liver in the mTOR regulation of lipid metabolism, insulin resistance, oxidative stress, and a series of other developmental processes during the circulation of the liver, causing liver inflammation, abnormal lipid metabolism, and other burdens. This will lead to a vicious circle of liver injuries and diseases, which then affect liver steatosis, fat accumulation, fibrosis, and other processes that lead to the occurrence and development of NAFLD. Genetic polymorphism and epigenetics are also closely related to OS and the regulation of the inflammatory response. Therefore, each influencing factor can affect NAFLD processes independently or through interactions with each other.

In conclusion, mTOR can not only regulate the developmental process of NAFLD via the molecular mechanism of the various influencing factors of NAFLD but also mediate the developmental trend of NAFLD by coordinating the relationship between various influencing factors of NAFLD. This is to prove the importance of mTOR in the influencing factors of NAFLD regarding multiple aspects.

## Figures and Tables

**Figure 1 ijms-23-09196-f001:**
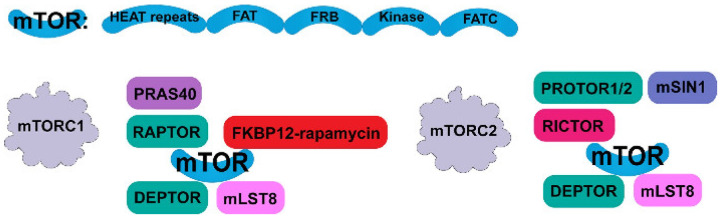
Architecture of mTORC1 and mTORC2. mTORC1 is mainly composed of mTOR, Raptor, mLST8, PRAS40, and DEPTOR. mTORC2 comprises mTOR, mLST8, Deptor, Rictor, mSIN1, and Protor1/2. mTOR is a complex protein composed of HEAT repeats and FAT, FRB, kinase, and FATC domains.

**Figure 2 ijms-23-09196-f002:**
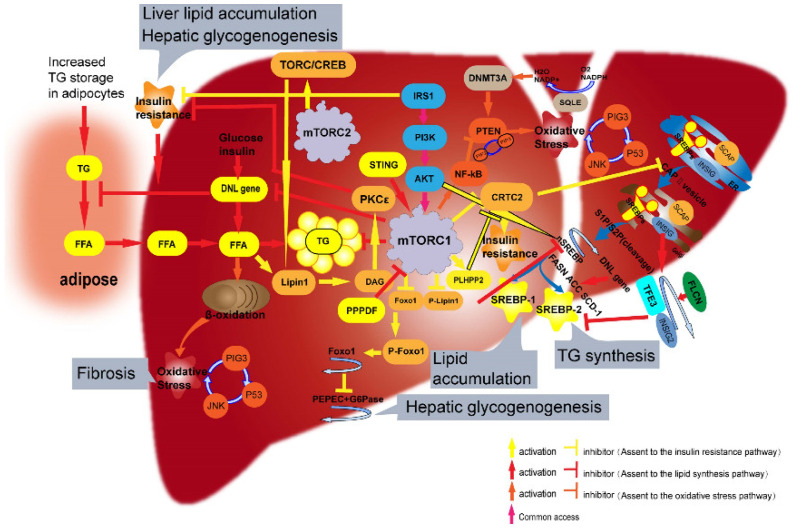
mTOR affects NAFLD through lipid metabolism (SREBPs), insulin resistance (Lipin1, Foxo1), and oxidative stress (PIG3, p53, JNK). (a). Excessive amounts of triglycerides stored in fat enter the liver as FFAs. Excessive triglyceride levels activate the feedback regulation of mTORC1 to block FFA-induced mitochondrial OS and the PIG3-p53-JNK-OS recycling pathway in the liver. mTORC1 can also regulate the effects of OS, mediated by NF-κB and PTEN. (b). FFAs activate Lipin1 and Foxo1 through mTORC1, thereby affecting DAG and PKC levels to influence hepatic IR. (c). mTORC1 can regulate SREBP-1 and SREBP-2 levels after activation by CRTC2 and can phosphorylate Lipin1, which affects insulin levels. Furthermore, mTORC1 can indirectly affect DNL-related gene levels by directly regulating the nSREBP gene via Akt. mTORC1 also directly suppresses DNL-related genes and thus exhibits feedback regulation of triglyceride synthesis. mTOR can affect triglyceride synthesis, liver lipid accumulation, liver gluconeogenesis, hepatitis, and fibrosis via the above three processes in NAFLD.

**Figure 3 ijms-23-09196-f003:**
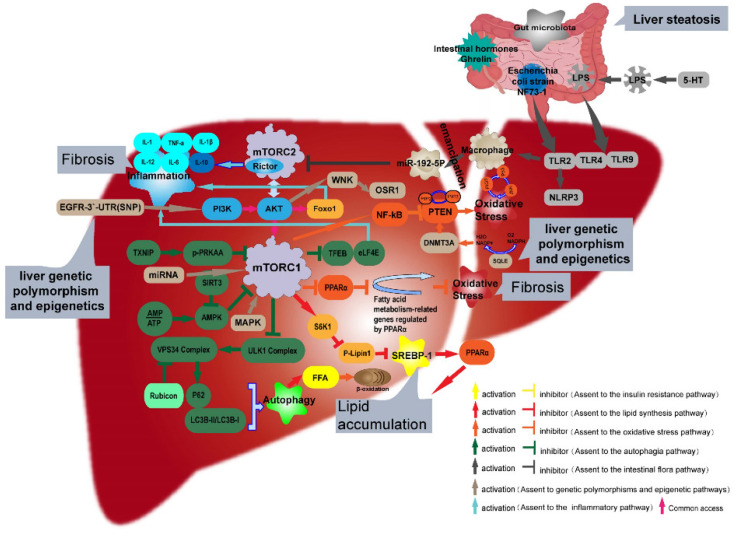
Diagram of the mechanism by which mTOR influences NAFLD based on the intestinal flora, autophagy, inflammation, genetic polymorphisms, and epigenetics. (a). LPS in the blood regulates the TLR family through the gut–liver axis, the intestinal flora and ghrelin, thus polarizing macrophages towards the proinflammatory M1 phenotype, inhibiting constitutively active Rictor in mTORC2 by releasing miR-192-5P from autophagic vesicles and ultimately affecting liver inflammation (via TNF-α, IL-1, IL-6, IL-10, etc.). (b). mTORC1, as the center of autophagy regulation, is closely associated with AMPK, TXINP, and the ULK1 and VPS34 complexes, and can affect liver inflammation through TFEB and eLF4E. (c). Akt and PI3K, which are upstream of mTORC1, are closely associated with genetic polymorphisms and epigenetic modification of WNK, OSR1, and EGFR-3′-UTR (SNP). DNMT3A and SQLE are key genes in the mTORC1-mediated NF-KB pathway that are regulated by genetic polymorphisms and epigenetics. (d). mTORC1 directly regulates S6K1 and phosphorylates Lipin-1 to inhibit SREBP-1 and influence PPAR, a key gene in OS. (e). mTORC1 can directly inhibit PPAR-α-mediated fatty acid metabolism and thus inhibit OS.

**Figure 4 ijms-23-09196-f004:**
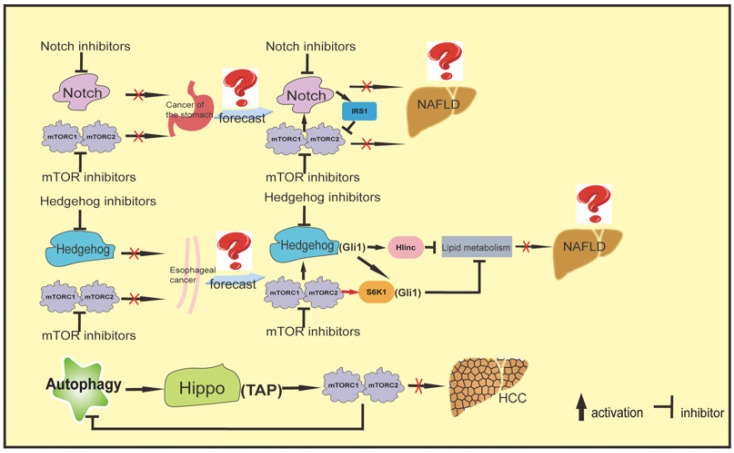
Crosstalk between mTOR and novel pathways: A source of potential new targets for the treatment of NAFLD-associated HCC. (a). Notch and mTOR synergistically promoted gastric cancer cell proliferation, while Notch or mTORC1 combined inhibition inhibited tumor cell growth. Hepatic Notch signaling increases insulin sensitivity by acting on IRS1 and decreases mTORC1-mediated SREBP-1c activity and hepatic triglycerides. However, the role of Notch and mTOR crosstalk in NAFLD remains unclear. The combination of (b). mTOR pathway inhibitors and Hedgehog pathway inhibitors inhibited esophageal cancer cells by up to 90%. The Hedgehog signaling pathway directly regulates Hilnc (Hedgehog signaling induced long non-coding Rnas) that participates in mTOR regulated liver lipid metabolism. Gli1 is identified as the substrate of S6K1, providing a Gli1 activation mechanism between mTOR and Hedgehog signaling pathways that is not dependent on Smoothened, a G-coupled receptor-like signal sensor. Therefore, it is speculated that crosstalk between the mTOR and Hedgehog signaling pathways can regulate NAFLD and even HCC through lipid metabolism. (c). YAP has a positive control effect on both mTORC1 and mTORC2, and the YAP protein can inhibit the initiation of autophagy and affect the progress of HCC by activating the mTORC1 signaling pathway, suggesting that the mTOR signaling pathway and Hippo pathway are crosstalk regulating HCC.

**Table 1 ijms-23-09196-t001:** Summary Table of mTOR-related signaling pathways affecting the various stages of NAFLD.

mTOR Related Signaling Pathways	The Stages That Affect NAFLD	Effects on the Course of Liver Disease	Reference
STING/mTORC1	NAFLD	Reduce fatty acid accumulation in hepatocytes	[33]
FLCN/mTORC1	NAFLD	Decreased lipid accumulation in the liver	[34]
mTORC1/SREBP1c/DNL	NAFLD	Liver lipid metabolism and fat deposition	[35,36,37,38]
AMPK/mTORC1	NAFLD	Reduce liver fat deposition; Mediating autophagy level and alleviating hepatotoxicity	[39,98,99,100]
IRS1/mTORC2/Foxo1	NAFLD	It regulates the occurrence of insulin resistance and affects hepatic glucose metabolism	[44]
Raptor(mTOR)/CRTC2	NAFLD	Inhibit hepatic steatosis; Regulates insulin secretion	[43,45,46]
mTORC2/CREB/LIPIN1	NAFLD	Regulation of LIPIN1 expression affects liver lipid accumulation	[47]
mTORC2/AKT/ATP-citrate lyase	NAFLD	Drives brown adipogenesis and de novo adipogenesis	[48]
mTOR/OS/JNK/p53	NAFLD and HCC	Regulating hepatocyte death affects autophagy	[51,52,53]
PI3K/AKT/mTOR	NAFLD, NASH and HCC	Induced oxidative stress and affected hepatocyte apoptosis; It affects the production of autophagy; Enhance autophagy flux, regulate autophagy and inhibit inflammation	[54,84,85,103,115,116]
PIG3/p53/OS/mTOR	NAFLD	Down-regulation or termination of insulin signaling downstream of PI3K affects cellular ROS level	[57,58,59]
PI3K/AKT/PTEN/mTOR	NAFLD	Cause OS to occur	[60,61,62,63,64]
LPS/TLR4,TLR9/mTOR	NAFLD and NASH	Activate the inflammatory cascade	[69,70,104,105]
p-AKT/mTOR/LC-3II	NAFLD	Macrophages were polarized toward a proinflammatory phenotype (M1)	[72]
mTOR/TLR2/NLRP3	NASH	It affects the production of M1 macrophages in liver	[73]
LPS/5-HT/mTOR	NAFLD	Improve intestinal barrier damage, alleviate liver inflammation	[74]
Ghrelin/AMPK/mTOR	NAFLD	It restored the up-regulation of autophagy and inhibited the translocation of NF-κB into the nucleus	[82]
mTORC2/AKT/Foxo1/3	HCC	Reduce the production of autophagy protein and inhibit autophagy	[81,82,83]
mTORC1/ULK1/ATG13	NAFLD	It affects the early stage of autophagy	[87,88,89]
Rubicon/mTOR	NAFLD and NASH	Accelerate hepatocyte apoptosis, excessive lipid accumulation and inhibit autophagy	[91,92]
mTOR/p70S6K, 4E-BP1 and eIF4E	NAFLD	Reduce hepatic lipid accumulation caused by inflammatory stress	[106,107,108]
SQLE/DNMT3A/PTEN/AKT/mTOR	NAFLD and HCC	Oxidative stress and reduced fat deposition in the liver	[65]
Osr1/PI3K/AKT/mTOR	NAFLD and NASH	Participates in the regulation of liver inflammation	[113]
microRNA/mTOR	NASH and HCC	Affect the occurrence of liver cancer	[114]
Notch/mTOR	NAFLD and HCC	Occur crosstalk	[117,118,119,120,121,122]
Hedgehog/mTOR	NAFLD and HCC	Occur crosstalk	[124,125,126,127,128]
Hippo/mTOR	NAFLD and HCC	Occur crosstalk	[104,133,134,135,136]

## Data Availability

Not applicable.

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
