# Peer review of "mTOR: A Potential New Target in Nonalcoholic Fatty Liver Disease"

_ijms, 2022, doi:10.3390/ijms23169196_

Round 1

Reviewer 1 Report

The review is indeed very interesting and well-written. Nevertheless, I would have the following suggestions:

- the authors should state also the limitations and streghts of this review

- the conclusions should be mentioned separately for clarity

Author Response

回复审稿人 1 条评论

亲爱的编辑和审稿人,

代表我的合著者,我们非常感谢您给我们机会来修改我们的手稿,我们非常感谢编辑和审稿人对我们题为“mTOR: A Potential New Target In非酒精性脂肪肝”(手稿 ID:ijms-1851452)。我们仔细研究了审稿人的意见,并在论文的“Track Changes”功能中进行了修改。我们想提交修改后的手稿。下面提供了对审稿人意见的答复。

对审稿人 1 评论的回应:

非常感谢您的建设性意见。

第 1 点: 1. 作者还应说明本综述的局限性和优势

回复1:非常感谢。对于我们的错误,我们深表歉意。以上问题请参考更新稿件,已在“12.讨论”中详细说明。

第 2 点: 2. 为清楚起见,应单独提及结论。

回复2:非常感谢。对于我们的错误,我们深表歉意。我们在更新的手稿中对其进行了修改,并分别描述了结论和展望。谢谢你。

Reviewer 2 Report

In this manuscript entitled ‘mTOR: A potential new target in nonalcoholic fatty liver disease’, the authors summarized mTOR-related pathways in on the occurrence and development of NAFLD-associated HCC. The detailed comments are listed below:

1.      Typo in line 45.

2.      The hypothesis needs to be strengthened in Introduction.

3.      Are there any therapeutic strategies targeting on mTOR? If so, please add it into the manuscript.

4.      The syntax of first sentence starting from line 695 is messy. (line 695-700).

Author Response

回复审稿人 2 条评论

亲爱的编辑和审稿人,

代表我的合著者,我们非常感谢您给我们机会来修改我们的手稿,我们非常感谢编辑和审稿人对我们题为“mTOR: A Potential New Target In非酒精性脂肪肝”(手稿 ID:ijms-1851452)。我们仔细研究了审稿人的意见,并在论文的“Track Changes”功能中进行了修改。我们想提交修改后的手稿。下面提供了对审稿人意见的答复。

对审稿人 2 评论的回应:

非常感谢您的建设性意见。

第1点: 1。1.第45行的错字。

回复1:非常感谢。对于我们的错误,我们深表歉意。至于“第45行的错字”这个问题,我们已经在手稿中更新了。谢谢你的纠正。“他”的意思是“NAFLD”。

第2点: 2.引言中的假设需要加强。

回复2:非常感谢。对于我们的错误,我们深表歉意。引言中的假设已被补充,细节反映在更正的手稿“引言”中。

Point 3: 3.有针对mTOR的治疗策略吗?如果是这样,请将其添加到手稿中。

回复3:感谢您的有用意见和建议。以上问题请参考更新稿件,已在“12.讨论”中详细说明。

Point 4: 4.从第695行开始的第一句语法混乱。(第 695-700 行)。

回复4:非常感谢。对于我们的错误,我们深表歉意。我们已经在更新的手稿中更正了它。谢谢你。

Reviewer 3 Report

Journal: IJMS                                                                           
Manuscript ID: ijms-1851452

Authors: Jiayao Feng et al.

Title: “mTOR: A Potential New Target In Nonalcoholic Fatty Liver Disease”

The authors of the present manuscript review the available data regarding the role of mTOR in NAFLD pathogenesis and the potential therapeutic implications of mTOR inhibition. It is an interesting and well-written review article.

A few minor comments:

1.     There are two reference lists. Please define which is correct.

2.     Please check the manuscript for potential typos (e.g., line 45: “he main influencing”)

3.     It would be helpful if the authors could add a summary table regarding the potential mTOR-related pathways/mechanisms for each stage of NAFLD (e.g., steatosis, NASH/fibrosis, etc.) and the effects of mTOR on lipid and glucose metabolism.

Author Response

回复审稿人 3 条评论

亲爱的编辑和审稿人,

代表我的合著者,我们非常感谢您给我们机会来修改我们的手稿,我们非常感谢编辑和审稿人对我们题为“mTOR: A Potential New Target In非酒精性脂肪肝”(手稿 ID:ijms-1851452)。我们仔细研究了审稿人的意见,并在论文的“Track Changes”功能中进行了修改。我们想提交修改后的手稿。下面提供了对审稿人意见的答复。

对审稿人 3 评论的回应:

非常感谢您的建设性意见。

Point 1: 1.有两个参考列表。请定义哪个是正确的。

回复1:非常感谢。对于我们的错误,我们深表歉意。至于“有两个参考列表。请定义哪个是正确的。”这个问题,我们已经在手稿中进行了更新。详情请参阅手稿中的“参考文献”。

第 2 点: 2.请检查稿件是否有潜在的拼写错误(例如,第 45 行:“他主要影响”)

回复2:非常感谢。对于我们的错误,我们深表歉意。关于这期“他主要郁闷”,我们已经在手稿中更新了。谢谢你的纠正。“他”的意思是“NAFLD”。

第 3 点: 3. 如果作者可以添加一个关于 NAFLD 每个阶段(例如,脂肪变性、NASH/纤维化等)的潜在 mTOR 相关途径/机制以及 mTOR 对脂质的影响的汇总表,将会很有帮助和葡萄糖代谢。

回复3:感谢您的有用意见和建议。对于上述问题,请参阅更新的手稿“表 1:影响 NAFLD 各个阶段的 mTOR 相关信号通路/机制汇总表”。
